# OpenReview forum: "INVITE: a Testbed of Automatically Generated Invalid Questions to Evaluate Large Language Models for Hallucinations"
_EMNLP/2023/Conference — EMNLP 2023 Findings_

### Official Review · Reviewer_5D6c · 2023-07-27

**Soundness:** 3

**Excitement:**

3: Ambivalent: It has merits (e.g., it reports state-of-the-art results, the idea is nice), but there are key weaknesses (e.g., it describes incremental work), and it can significantly benefit from another round of revision. However, I won't object to accepting it if my co-reviewers champion it.

**Missing References:**

Missing reference about Question Generation field.
such as:
1. Question generation for question answering
2. A review on question generation from natural language text

**Paper Topic And Main Contributions:**

The paper proposes a tested of automatically generating invalid questions to evaluate LLMs' hallucinations. The contribution can be concluded:
1. They propose a framework to create the invalid questions.
2. They test the framework and conduct a pilot human evaluation on the generated respones.

**Reasons To Accept:**

1. The paper is well written and easy to understand.
2. There are many types of questions in the framework, which could make the testbed more robust.
3. The paper used multiple LLMs for generating response and human evaluation.

**Reasons To Reject:**

1. Since the Paper focused on evaluating LLMs' hallucination problem, it's better to also conduct experiments on larger LLMs such as GPT3.5 or GPT4.
2. The question generation approach itself is very simple (fill the entities to fixed template) and the author only use one template for each question category. The diversity of the generated test set is limited, which could cause bias on evaluation.
3. The paper used generac automatic evaluation metrics that not for evaluating hallucination, which doesn't add any valueable information for the paper. Maybe use some more metrics that specific for hallucination evaluation.

**Reproducibility:**

5: Could easily reproduce the results.

**Reviewer Confidence:**

4: Quite sure. I tried to check the important points carefully. It's unlikely, though conceivable, that I missed something that should affect my ratings.

---

> ### Author Rebuttal · Authors · 2023-08-29
>
> Thank you for your feedback. Please find answers to your questions below:
>
> * *Since the Paper focused on evaluating LLMs' hallucination problem, it's better to also conduct experiments on larger LLMs such as GPT3.5 or GPT4*
>     * We conducted additional experiments where we obtained labels from both the GPT3.5 and GPT4 models from OpenAI which we will also add to the final draft.
>         * The hallucination rates from both models are as follows:
>             * GPT3.5: 21.5% of responses were hallucinated.
>             * GPT4: 6.7% of responses were hallucinated.
>         * While both models showed reduced rates of hallucinations, they also showed considerably higher disengagements even with valid questions from our testbed:
>             * GPT3.5: 26% of all responses to valid questions were disengagements.
>             * GPT4: 29% of all responses  to valid questions were disengagements.
>         * As noted earlier, the GPT3.5 and GPT4 models seem to fare well with our synthetically created dataset. Further, since OpenAI does not release internal details of these models, it is unclear if these models incorporate any form of external knowledge retrieval which would provide an effective solution to handle any question with invalid assumptions such as those presented in this paper. However, given the growing popularity of LLMs and the demand to deploy them in various low latency regimes, we argue that our approach still provides valuable benefits in evaluating smaller (and ungrounded) models to measure their tendency for hallucinations. Further, even identifying the 6.7% use cases where GPT-4 hallucinates highlights existing gaps in these models and provides valuable insights needed to make these models safe for deployment.
> * *The question generation approach itself is very simple (fill the entities to fixed template) and the author only use one template for each question category. The diversity of the generated test set is limited, which could cause bias on evaluation.*
>     * We note that our current framework covers 24 different domains spanning a variety of entities and relationships (such as award, successor, etc.) and hence already provides a fairly robust coverage in evaluating the models’ tendency to hallucinate. We believe this to be a simple yet promising early finding direction for LLM evaluations. However, we acknowledge that within each domain, we currently use a single question but note that we are currently expanding these further by collecting at least 10-20 additional templated questions for each domain using crowdsourcing, which will further be expanded using automated paraphrasing (we will include these results in the camera ready draft if it is permitted). In a future update we will also explore constrained question generation strategies similar to [Duan et al. 2017].
> * *The paper used generac automatic evaluation metrics that not for evaluating hallucination, which doesn't add any valueable information for the paper. Maybe use some more metrics that specific for hallucination evaluation.*
>     * We will add a recently released automated metric titled AlignScore [Zhu et al. 2023] in the final draft. This metric has been shown to correlate well with human evaluations of hallucinations (see Table 4 from [Zhu et al. 2023]).
> * *Missing reference about Question Generation field*
>     * We will add these references to the camera ready draft.
>
> [Duan et al, 2017] Question Generation for Question Answering
>
> [Zhu et al. 2023] ALIGNSCORE: Evaluating Factual Consistency with A Unified Alignment Function

---

### Official Review · Reviewer_obFN · 2023-08-04

**Soundness:** 3

**Excitement:**

4: Strong: This paper deepens the understanding of some phenomenon or lowers the barriers to an existing research direction.

**Paper Topic And Main Contributions:**

This paper introduces INVITE, a testbed for evaluating LLM hallucination. This testbed is generated automatically and tries to fool LLMs by posing a question with incorrect assumptions (e.g., when did A start education at B, where A has never studied at B). To generate the questions, they use the 100 most frequent DBpedia predicates (noisy predicates are discarded). They then come up with predicate specific templates using a search engine (to make sure the SE returns unambiguous responses). They use a valid triple (given a target predicate) to write a question. Then, they perturb the subject or the object to create an invalid question. They also generate questions with invalid dates.

They come up with 150 of such invalid questions, where their responses (from 5 open-source LLMs up to 13B parameters) are manually evaluated. They show a high rate of hallucination (>=72.9% where they don't consider empty responses as hallucination). This shows that LLMs can be easily fooled with invalid questions. They also perform automatic evaluation, but don't see a meaningful correlation with human evaluation.

**Reasons To Accept:**

1. The fact that LLMs can be easily fooled with invalid questions is important and it is shown in the paper. The LLMs should improve so that they work reliably even with noisy input.

2. A wide range of LLMs are tested to support the idea.

3. The reliance on templates and DBpedia sounds like a reasonable approach to build such a dataset.

**Reasons To Reject:**

1. They don't clarify if they will actually release the dataset. I understand that the dataset can be generated again with their codebase, but it would be useful to have the dataset ready to quickly run experiments.

2. The automatic evaluation doesn't correlate well with human evaluation. This makes the dataset much harder to use. I wonder if a better automatic metric could help (e.g., NLI-based metrics).


**Reproducibility:**

4: Could mostly reproduce the results, but there may be some variation because of sample variance or minor variations in their interpretation of the protocol or method.

**Reviewer Confidence:**

4: Quite sure. I tried to check the important points carefully. It's unlikely, though conceivable, that I missed something that should affect my ratings.

---

> ### Author Rebuttal · Authors · 2023-08-29
>
> Thank you for the positive review, we appreciate your feedback. Please find our answers to your questions below:
>
> * *They don't clarify if they will actually release the dataset. I understand that the dataset can be generated again with their codebase, but it would be useful to have the dataset ready to quickly run experiments.*
>     * Yes, we will release our final dataset of questions along with the corresponding predictions from all the models we evaluate and human labels (we will add this note to our draft as well).
> * *The automatic evaluation doesn't correlate well with human evaluation. This makes the dataset much harder to use. I wonder if a better automatic metric could help (e.g., NLI-based metrics).*
>     * In addition to the metrics listed in the paper, in the final draft of the paper, we will also include evaluations from a recently released automated metric titled AlignScore [Zhu et al. 2023]. This metric has been shown to correlate well with human evaluations of hallucinations (see Table 4 from [Zhu et al. 2023]).
>
> [Zhu et al. 2023] ALIGNSCORE: Evaluating Factual Consistency with A Unified Alignment Function

---

### Official Review · Reviewer_E6J6 · 2023-08-05

**Soundness:** 3

**Excitement:**

2: Mediocre: This paper makes marginal contributions (vs non-contemporaneous work), so I would rather not see it in the conference.

**Paper Topic And Main Contributions:**

The paper presents a new framework called INVITE, which stands for a testbed of automatically generated invalid questions to evaluate large language models for hallucinations.The framework creates invalid questions by distorting valid facts in which subjects or objects are replaced by similar entities. Several state-of-the-art large language models are evaluated against a generated test set generated by the INVITE and highlights its capacity to trigger hallucinations in these models.
Main Contributions:
1. A new framework for generating invalid questions to evaluate LLMs for hallucinations.
2. The evaluation of existing LLMs against the hallucination-trigger test set
3. The proposal of an alternative strategy to evaluate LLMs on hallucinations using adversarial questions with invalid assumptions.

**Questions For The Authors:**

1. What is the essential difference between INVITE and existing knowledge conflict approaches, and will your strategy provide a better reflection to large language model hallucinations?
2. The experiment only tested the effect on a few pre-trained and SFT models, what happens if a larger or aligned model is used for the experiment.
3. It appears that only one question template was used to test the model on a particular domain. How variance this method is when the template is modified, and whether the model gave different feedback for different question-answer formats.

**Reasons To Accept:**

1. The research idea is straightforward, intuitive and rational.
2. A validation verifies the excitation of large language model hallucinations by misinformation in the context.

**Reasons To Reject:**

1. The research idea is not novel, and the methodology is simply a change in the prompting instrustion relative to existing trustworthy QA datasets.
2. Experimental results are limited and should be tested simultaneously on larger language models

**Reproducibility:**

5: Could easily reproduce the results.

**Reviewer Confidence:**

4: Quite sure. I tried to check the important points carefully. It's unlikely, though conceivable, that I missed something that should affect my ratings.

---

> ### Author Rebuttal · Authors · 2023-08-29
>
> Thank you for your feedback. Please find answers to your questions below:
>
> * *What is the essential difference between INVITE and existing knowledge conflict approaches, and will your strategy provide a better reflection to large language model hallucinations?*
>     * The key difference between our approach and approaches using knowledge conflicts (such as Longpre, et al. 2021, Schuster et al., 2021, etc.) is that in these works, the authors first start with an available tuple of valid question, context and answers, from a suitable corpus where they replace all occurrences of the answer with a newly sampled entity. However, the main premise of the original question/context remains unaffected, which often contains sufficient context for most modern LLMs to disambiguate the added conflict. In contrast, in our work we use a templated question which primarily includes the conflicting fact and little to no additional context.
>         * Consider this for example from [Longpre et al., 2021]: “*The United States declared war on Germany on April 6, 1917, over 2 years after World War I started*”; given this, this work samples a new entity Taiwan and replaces every occurrence of Germany with Taiwan, leading to this distorted question: “*The United States declared war on Taiwan on April 6, 1917, over 2 years after World War I started*”. While [Longpre et al.] use this approach only to distort a context, it is straightforward to extend this approach to distort questions.
>     * However, the context still has sufficient information (such as the exact date or this declaration and the fact that this was in World War I) for an LLM to unambiguously answer the question without any need for hallucinations. In contrast, our approach creates a new set of questions with little or no valid context which makes it easier to draw out the tendency for hallucinations in LLMs.
> * *The experiment only tested the effect on a few pre-trained and SFT models, what happens if a larger or aligned model is used for the experiment.*
>     * To supplement the results presented in the paper, we conducted additional experiments where we obtained labels from GPT3.5 and GPT4 models from OpenAI, which we will add to the final draft.
>         * The hallucination rates from both models are as follows:
>             * GPT3.5: 21.5% of responses were hallucinated.
>             * GPT4: 6.7% of responses were hallucinated.
>         * While both models showed reduced rates of hallucinations, they also showed considerably higher disengagements even with valid questions from our testbed:
>             * GPT3.5: 26% of all responses to valid questions were disengagements.
>             * GPT4: 29% of all responses  to valid questions were disengagements.
>     * As observed in a variety of other recent evaluations, the GPT3.5 and GPT4 models seem to fare well with our synthetically created dataset. Further, since OpenAI does not release internal details of these models, it is unclear if these models incorporate any form of external knowledge retrieval which would provide an effective solution to handle any question with invalid assumptions such as those presented in this paper. However, given the growing popularity of LLMs and the demand to deploy them in various low latency regimes, we argue that our approach still provides valuable benefits in evaluating smaller (and ungrounded) models to measure their tendency for hallucinations. Further, even identifying the 6.7% use cases where GPT-4 hallucinates highlights existing gaps in these models and provides valuable insights needed to make these models safe for deployment.
> * *It appears that only one question template was used to test the model on a particular domain. How variance this method is when the template is modified, and whether the model gave different feedback for different question-answer formats.*
>     * We note that our current framework covers 24 different domains spanning a variety of entities and relationships (such as award, successor, etc.) and hence already provides a fairly robust coverage in evaluating the models’ tendency to hallucinate. We believe this to be a simple yet promising early finding direction for LLM evaluations. However, we acknowledge that within each domain, we use a single question but note that we are currently expanding these further by collecting at least 10-20 additional templated questions for each domain using crowdsourcing, which will further be expanded using automated paraphrasing (we will include these results in the camera ready draft if it is permitted).
>
> [Longpre, et al. 2021] Entity-Based Knowledge Conflicts in Question Answering
>
> [Schuster et al., 2021] Get Your Vitamin C! Robust Fact Verification with Contrastive Evidence

---

### Meta-Review · Senior_Area_Chairs · 2023-10-04

**Recommendation:** 3

**Metareview:**

I provide here an analysis of three reviews for the paper. The reviews present both positive and negative aspects of the paper, leading to a comprehensive evaluation of its quality.

Reasons to Accept:

Importance of the Research Problem: The paper addresses an important issue in the field of large language models (LLMs) by investigating their susceptibility to hallucinations when confronted with invalid questions. This problem is significant as it highlights the need for improving the reliability of LLMs.

Diverse LLM Evaluation: The paper tests a wide range of LLMs, which strengthens its claims and provides a comprehensive evaluation of the problem.

Reasonable Data Generation Approach: The use of templates and DBpedia predicates to generate invalid questions is considered a reasonable approach, providing a basis for creating a valuable dataset for future research.
Clarity and Accessibility: Reviewers appreciate the paper's clear and understandable presentation, making it accessible to a broad audience.

Reproducibility: The paper's methodology is deemed reproducible, enhancing the credibility of the research.

Reasons to Reject:

Lack of Dataset Release Clarity: Reviewers express concerns about the unclear status of dataset release. While the dataset can potentially be generated with the provided code, clarity on dataset availability would facilitate quicker experimentation.

Poor Correlation Between Automatic and Human Evaluation: The paper reports a poor correlation between automatic and human evaluation, which raises questions about the utility of the dataset. Suggestions are made to explore better automatic metrics for evaluation.

Limited Question Generation Diversity: The simplicity of the question generation approach, where entities are filled into fixed templates, and the use of only one template per question category, are seen as limitations that may introduce bias into the evaluation.

Exclusion of Larger LLMs: The paper focuses on smaller LLMs for evaluation and does not include experiments on larger models like GPT-3.5 or GPT-4, which could provide more insights into the problem. Additional Metrics for Hallucination Evaluation: Reviewers suggest incorporating specific metrics tailored for hallucination evaluation to provide more valuable insights into LLM performance.

---

### Decision · Program_Chairs · 2023-10-07

**Decision:**

Accept-Findings

**Comment:**

I provide here an analysis of three reviews for the paper. The reviews present both positive and negative aspects of the paper, leading to a comprehensive evaluation of its quality.

Reasons to Accept:

Importance of the Research Problem: The paper addresses an important issue in the field of large language models (LLMs) by investigating their susceptibility to hallucinations when confronted with invalid questions. This problem is significant as it highlights the need for improving the reliability of LLMs.

Diverse LLM Evaluation: The paper tests a wide range of LLMs, which strengthens its claims and provides a comprehensive evaluation of the problem.

Reasonable Data Generation Approach: The use of templates and DBpedia predicates to generate invalid questions is considered a reasonable approach, providing a basis for creating a valuable dataset for future research.
Clarity and Accessibility: Reviewers appreciate the paper's clear and understandable presentation, making it accessible to a broad audience.

Reproducibility: The paper's methodology is deemed reproducible, enhancing the credibility of the research.

Reasons to Reject:

Lack of Dataset Release Clarity: Reviewers express concerns about the unclear status of dataset release. While the dataset can potentially be generated with the provided code, clarity on dataset availability would facilitate quicker experimentation.

Poor Correlation Between Automatic and Human Evaluation: The paper reports a poor correlation between automatic and human evaluation, which raises questions about the utility of the dataset. Suggestions are made to explore better automatic metrics for evaluation.

Limited Question Generation Diversity: The simplicity of the question generation approach, where entities are filled into fixed templates, and the use of only one template per question category, are seen as limitations that may introduce bias into the evaluation.

Exclusion of Larger LLMs: The paper focuses on smaller LLMs for evaluation and does not include experiments on larger models like GPT-3.5 or GPT-4, which could provide more insights into the problem. Additional Metrics for Hallucination Evaluation: Reviewers suggest incorporating specific metrics tailored for hallucination evaluation to provide more valuable insights into LLM performance.